# Earlier Diagnosis of Pancreatic Cancer: Is It Possible?

**DOI:** 10.3390/cancers15184430

**Published:** 2023-09-05

**Authors:** Tomas Koltai

**Affiliations:** Hospital del Centro Gallego de Buenos Aires, Buenos Aires C1094, Argentina; tkoltai@hotmail.com

**Keywords:** pancreatic cancer, screening, endoscopic ultrasound, tumor markers, natural history of pancreatic cancer, early diagnosis, intraductal neoplasia

## Abstract

**Simple Summary:**

Pancreatic cancer incidence is increasing yearly. The reasons are not well known. Unfortunately, this is one of the least treatable cancers. Standard chemotherapy treatments show poor results, as do targeted treatments. The only real improvement in pancreatic cancer in the last twenty years occurred in the surgical field, where neoadjuvant therapy and very early surgery have achieved better overall survival. The only secret of arriving early to surgery is early diagnosis, and the missing element for early diagnosis is screening. This paper discusses the population that needs to be screened.

**Abstract:**

Pancreatic ductal adenocarcinoma has a very high mortality rate which has been only minimally improved in the last 30 years. This high mortality is closely related to late diagnosis, which is usually made when the tumor is large and has extensively infiltrated neighboring tissues or distant metastases are already present. This is a paradoxical situation for a tumor that requires nearly 15 years to develop since the first founding mutation. Response to chemotherapy under such late circumstances is poor, resistance is frequent, and prolongation of survival is almost negligible. Early surgery has been, and still is, the only approach with a slightly better outcome. Unfortunately, the relapse percentage after surgery is still very high. In fact, early surgery clearly requires early diagnosis. Despite all the advances in diagnostic methods, the available tools for improving these results are scarce. Serum tumor markers permit a late diagnosis, but their contribution to an improved therapeutic result is very limited. On the other hand, effective screening methods for high-risk populations have not been fully developed as yet. This paper discusses the difficulties of early diagnosis, evaluates whether the available diagnostic tools are adequate, and proposes some simple and not-so-simple measures to improve it.

## 1. Introduction

Pancreatic cancer (PC) is becoming a public health problem because the number of cases is constantly increasing [1], and treatment results are quite poor [2,3,4]. Incidence is growing between 0.5 and 1% each year. In 1985, PC was the eighth most frequent cause of cancer mortality [5]. Thirty years later, it was the fourth [6], but it has been forecast that it will be in second place by 2030–2040 [7,8]. In October 2022, we found over 3300 completed or ongoing clinical trials on the clinicaltrials.gov webpage, which indirectly shows the magnitude of the problem.

The most frequent tumor is the pancreatic ductal adenocarcinoma (PDAC), which represents 90% of pancreatic malignancies [9,10]. From here on, unless otherwise stated, this paper will be limited to PDAC.

One important issue is that in spite of all the recent advances in knowledge about PDAC, we still do not fully understand its biology.

Surgery is still the best treatment choice, which offers the highest 5-year survival [11,12]. However, the proportion of operable patients is low and ranges between 15 and 20% [13].

The number of operable cases increased in the last 20 years due to the introduction of neoadjuvant chemotherapy to reduce tumor size in locally advanced PDAC [14,15]; it also seems to improve overall survival [16]. On the other hand, when tumors are small, less than 2 cm. stage T1N0M0, 5-year survival increases substantially [17,18].

When tumors are less than 1 cm, a 5-year survival above 50% can be achieved [19].

Most patients are diagnosed in a late stage when surgery is not feasible, or metastases are already present. Symptomatic patients are usually incurable [20]. At the time of diagnosis in 80 to 85% of patients, an invasive pancreatic tumor of 4 cm or more in diameter is present. In most of these cases, there are overt or occult metastases as well [21,22].

Inoperable patients can be treated with chemotherapy and radiotherapy; however their response rate is low, and survival time is short. Targeted treatments, such as poly (adenosine diphosphate [ADB]-ribose) polymerase inhibitors, are limited to a small subset of patients with BRCA1/BRCA2 mutations.

Furthermore, the 5-year survival with this cancer is negligible. Until the beginning of the twenty-first century, 95% of patients with PC succumbed within two years [23]. PC is the malignancy with the shortest survival among gastrointestinal cancers [24].

These unfortunate results have improved only slightly in the last ten years (the 5-year survival rate was 5.26% in 2000 and increased to nearly 10% in 2020 [25]). The proportion of patients diagnosed in stage IA has slightly increased, and this may be the result of an earlier diagnosis [26].

Therefore, a reliable screening method is undoubtedly needed. The best proof of this disease’s lethality is that the incidence rate is very close to its mortality rate.

The poor outcome in PC is mainly due to five characteristics of these tumors [27]:(1)aggressive biological behavior;(2)extensive invasion;(3)lack of early specific symptoms and thus delayed diagnosis;(4)dense stroma that participates in the cancer’s progression and impedes the delivery of chemotherapy drugs to the tumor;(5)early and frequent development of multidrug resistance [28].

## 2. Natural History of Pancreatic Cancer

There is evidence that the symptomatic disease takes around fifteen years to develop from when the first pro-tumor genotypic change occurs [29]. PDAC development follows a step-by-step process from intraepithelial dysplasia and intraductal neoplasia to full-blown invasive adenocarcinoma [30,31]. There is a series of precursor lesions, such as intraepithelial neoplasia and intraductal papillary mucinous tumors [32] (the Sendai and Fukuoca protocols established when these cysts should be removed) [33,34].

Distant metastases also develop late [26]. This evidence shows that contrary to what most textbooks say, PDAC is a slowly developing cancer. What probably creates the contradiction is that by the time it is diagnosed—usually late—the tumor has achieved cumulative mutations that accelerate its progression, and very frequently, distant metastases have already occurred. The time interval between early and late diagnosis is less than 1½ years (Figure 1).

Intraepithelial neoplasms should be considered precancerous lesions. However, it is not clear which will progress to PDAC and which will remain unchanged [35]. The KRAS mutation is insufficient for distinguishing those that will follow a malignant path because both can show this mutation [36,37]. Furthermore, only 75% of PCs have a KRAS mutation. Probably finding coexistent KRAS and TP53 mutations is strong evidence for a malignant evolution. This needs further confirmation.

On the other hand, the KRAS mutation is an important hallmark that helps rule out chronic pancreatitis [38].

**Figure 1 cancers-15-04430-f001:**
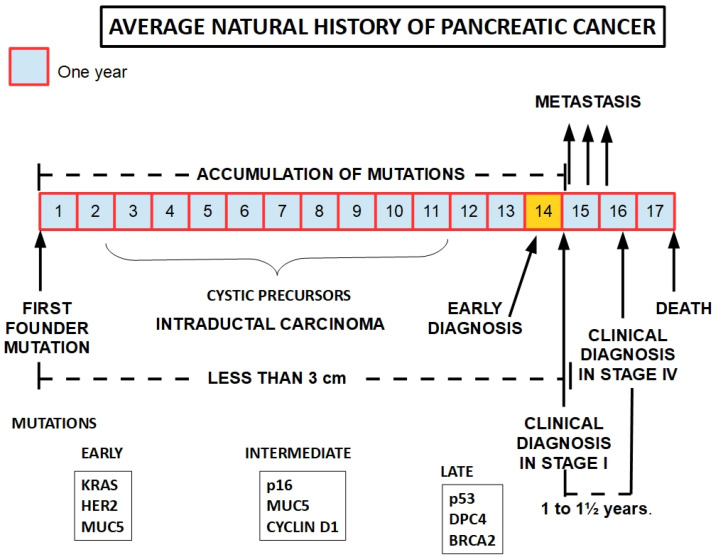
The development of pancreatic cancer is a slow process that takes around 15 years to become symptomatic. The pace of progression seems to accelerate concomitantly with the presentation of the first symptoms. In order to be therapeutically effective, early diagnosis needs to occur immediately before symptoms develop or very shortly afterwards. This therapeutic window may be as short as weeks or a few months at best. Lower panels show frequently found mutations according to the progression of the tumor [39].

## 3. Growth of Pancreatic Cancer

The classic tool, although inaccurate, for measuring tumor growth is tumor volume duplication time (TVDT). TVDT is usually determined from two-volume estimations with measurement time intervals and establishes the time it takes to double the volume of a tumor. Unfortunately, there are many publications with very different results. Therefore, the information gathered in this regard should be considered with prudence.

According to Kay et al. [40], extracting doubling times from progression-free survival (PFS) plots showed that pancreatic cancer doubles its volume in 5.3 months. If we compare this growth with other tumors, we have to conclude that pancreatic cancer does not have a particularly accelerated growth. Examples obtained from the same publication by Kay et al.:
Melanoma3.78 monthsHepatocarcinoma3.06 monthsRenal cell carcinoma2.67 monthsTriple negative breast cancer2.38 monthsNon-small cell lung cancer2.40 monthsHormone-positive breast cancer4.31 monthsHer 2 positive breast cancer4.12 monthsGastric cancer3.82 monthsGlioblastoma2.55 monthsProstate4.10 months

Ahn et al. [41] studied duplication time with CT scan in 110 patients with proven PDAC and found a very wide range that went from 20 to 977 days (mean 132 days): “*The growth rate was significantly associated with the initial diameter and volume. The development of distant metastasis was significantly associated with initial diameter, volume, and volume growth rate”*.

Furukawa et al. [42] found a much narrower range between 64 and 255 days, but with a similar average to that of Kay, 144 days.

If we use 5 months as an average for PC duplication time and the best results obtained by surgery when the tumor has a diameter of less than 3 cm, it is easy to understand that the therapeutic window for achieving good surgical results is very small, probably much smaller than the one shown in Figure 1.

## 4. A Growth Model of Pancreatic Cancer

Although not universally accepted, a growth model for pancreatic cancer progression can be built following the multistep scheme that Vogelstein, Feron, and Kinslay proposed for colorectal carcinoma [43,44].

In the pancreas, the initial lesion is probably a mutation of an intraductal cell, the founder mutation, leading to intraductal proliferation or intraductal hyperplasia [45]. It is this dysplastic duct that evolves into the invasive adenocarcinoma through clonal evolution [46,47]. These findings were also corroborated on clinical grounds [48,49] and by genetic research [50,51,52,53,54]. For example, the KRAS mutation, the most frequently found driver mutation of PDAC, has also been frequently found in intraductal carcinoma [55,56]. Other shared genetic alterations are Ras and BRCA2 mutations and loss of p53 and p16 [57,58,59,60]. These mutations accumulate over time, and this is the possible reason why sporadic PC appears mainly in patients over 50 years of age.

## 5. Precursor Lesions

If we want to change the present situation of late diagnosis, it is imperative to look into the precursor lesions that precede the highly invasive characteristics of PDAC by many years. Any of the different types of precursor lesions can develop into PDAC.

The precursor lesions of pancreatic cancer can be divided into the following groups [61,62]:**Pancreatic intraepithelial neoplasias (PanIN),** which are usually flat, and importantly, they are non-invasive. They are classified into four categories based on the degree of dysplasia [63]: PanIN-1A, PanIN-1B, PanIN-2, and PanIN-3.**Intraductal papillary mucinous neoplasms (IPMN)**, which are usually large, mucin-producing epithelial lesions originating from the main pancreatic duct or major branch ducts. Their size, which is bigger than PanIN precursor lesions, makes them easy to detect by conventional imaging techniques.

PanIN and IPMN are considered neoplasias. However, both can be subdivided into low-grade and high-grade dysplasia. While this last subgroup is usually surgically resected, the first group is amenable to follow-up observation [64].

**Mucinous cystic neoplasm (MCN),** which also includes two subgroups, low and high grade.

The classification and progression scheme we have described above, starting with low-grade PanIn, high-grade PanIn, low-grade IPNM, high-grade IPNM, low-grade MCN and high-grade MCN, has direct implications for early diagnosis and has been summarized in a seminal publication by Hruban et al. [65] and confirmed by genetic findings by Notta et al. [66]. These implications are:(1)The aim of early diagnosis should focus on discovering these precursor neoplasias; this means that early diagnosis should achieve diagnosis even before a small invasive tumor develops.(2)Intraductal neoplasia has genetic alterations that in many cases can be detected in stool and pancreatic juice.(3)Removal of high-grade lesions in this “ultra” early stage would fundamentally change patient outcomes.(4)Present-day methods, whether biomarkers or imaging techniques, are ineffective in many of these precursor lesions.(5)Following the words of Hruban et al. [65] “*This progression model suggests that these early pancreatic duct lesions in the pancreas might also be reasonable targets for chemoprevention. For example, the progression model for colorectal carcinoma has formed the basis for chemoprevention trials in patients with familial adenomatous polyposis. Similarly, patients with an inherited susceptibility to pancreatic cancer may also be a reasonable group to study the benefit of chemoprevention of pancreatic cancer*”.

## 6. Early Diagnosis

Based on the issues discussed above, we need to redefine the concept of early diagnosis. The classic description sets two conditions:(1)a tumor with a diameter of less than three centimeters that(2)has not metastasized.

To these two conditions, we have added a third: lack of involvement of critical vessels.

The metastasis criteria are based on clinical grounds, and it would not be impossible for sub-clinical microscopic metastasis to be present.

The redefinition of early diagnosis consists of achieving a diagnosis before the neoplasia becomes invasive. Following the criteria of the International Cancer of the Pancreas Consortium, early diagnosis “should detect and treat T1N0M0 margin-negative PC and high-grade dysplastic precursor lesions” [67].

We shall analyze the feasibility below.

## 7. Diagnosis and Surgical Treatment

The clinical diagnosis of pancreatic cancer at an early stage is usually difficult because:(a)the disease is asymptomatic in the early stage;(b)the organ is hidden in the retroperitoneum;(c)there are no reliable early tumor markers;(d)the existing markers are not sufficiently specific to differentiate benign from malignant disease;(e)imaging techniques do not always allow the diagnosis of small surgically resectable cancers, and they are expensive;(f)pre-invasive neoplasias are frequently beyond the abilities of imaging techniques, and usual biomarkers are not increased.

Classic and new targeted treatments have almost no impact on pancreatic cancer outcomes, while early diagnosis, when surgery is possible, shows a significantly improved overall survival. Approximately 20% of patients in which surgery is possible will be alive after 5 years [11]. The percentage is somewhat higher in patients with a tumor of less than 3 cm in diameter. These findings fully justify the search for early diagnostic methods. If the diagnosis can be achieved before the invasive stage, the results should improve substantially.

Mortality and morbidity of classical duodenopancreatectomy (Whipple’s procedure) have decreased in the last 15 years, and overall survival has also improved in surgical cases. Neoadjuvant treatments have increased the rate of operable PDACs [68,69]. This background justifies the search for a reliable screening method leading to what here we have called the “ultra” early detection. However, there is an important step before screening: determining whom to screen.

Due to the need for imaging studies in PC it is almost inevitable to include them in any screening method. Their cost is high, and therefore the population to be screened should be limited to high-risk individuals.

There is no reliable early marker for PC, although there are some in more advanced disease and prognostic markers. The need for late and prognostic markers may be important for treatment follow up, but they are of little help if we want to screen large at- risk populations at an early stage.

This paper will propose an algorithm to identify high-risk populations and a screening method oriented to improving early diagnosis, which is essential if we want a better clinical outcome.

The first step in this quest should be the identification of the population at risk. The second step should be the screening of this population with some low-cost method, and finally, after refining the search to those cases that seem to be at very high risk, imaging studies such as MRI and endoscopic ultrasound can represent the final step.

This paper is divided into two parts. In the first part, PC risk is analyzed, and a scoring method is proposed. The second part discusses possible screening methods.

## 8. Part 1: Pancreatic Cancer Risk

### 8.1. Population at Risk

Due to economic burdens and medical availability, it is probably impossible to screen populations indiscriminately. On the other hand, identifying a population with a higher risk for pancreatic cancer and limiting the screening to this group seems a feasible approach.

### 8.2. Determining Population at Risk

**Age**: The first risk factor to analyze should be age. It is highly infrequent to find pancreatic cancer in people under 50 [70]. The median age at diagnosis in the white population is 75 years [71]. The median age in the USA is 71 years [72]. However, native African populations showed a median age of 55.7 years, while African Americans showed a median age of 66.7 [73,74]. Pancreatic cancer occurs at a younger age in native Africans when compared to the African American and white populations. In familial pancreatic cancer, it is frequently found that the tumor appears at a younger age than in sporadic cases [75].

**Race:** African Americans have a 20% higher risk than the white population [76].

**Chronic pancreatitis:** Chronic pancreatitis represents an important increase in the risk for pancreatic cancer. According to a meta-analysis of large population groups, Kirkegård et al. [77] found that the risk increase was 8-fold when compared with equivalent normal populations. This risk increase is independent of sex, country, and type of pancreatitis [78].

**Hereditary factors:** There is a small group of patients, approximately 10%, in which many cases can be identified in the same family: family clustering. Individuals that belong to families in which two or more cases were identified have a higher risk for PC [79,80,81,82,83,84,85,86,87,88,89]. There is evidence that these cases have a genetic base [90]. The PALB2 gene was discovered to be a familial pancreatic cancer susceptibility gene [91]. This gene is a binding partner of BRCA2. Mutations of BRCA2 and PALB2 are the most frequently found genetic alterations in familial PC.

**Cigarette smoking:** Most publications agree that cigarette smoking represents an important risk factor for PDAC [92,93,94,95,96,97]. According to Silverman et al. [98], eliminating cigarette smoking would decrease pancreatic cancer incidence by 27% in the United States. In northern Italy, this figure seems to be somewhat lower: 13.6% [99].

In a large population study, Boyle et al. [100] determined that the risk of PC increased with increasing lifetime consumption of cigarettes, with the relative risk reaching 2.70 in the highest smoking group. Most publications also agree on higher risk with longer smoking duration, higher amount, and higher cumulative dose.

Bertuccio et al. [101] found that cigar smoking was also associated with an excess risk of pancreatic cancer. This was not the case for pipe smoking and smokeless tobacco use. Smokers of black tobacco are at a higher risk than blond tobacco smokers (OR 2.09 vs. OR 1.43, respectively) [102].

Occasional smokers and those who smoke less than 10 cigarettes a day show no risk increase [93].

Smoking cessation is probably the only available preventive measure for pancreatic cancer [103]. However, there are some publications that did not find a correlation between smoking and PDAC [104]. The mechanisms involved have not been clearly established as yet [105]. Furthermore, smoking has been found to be related to reduced survival in patients with pancreatic cancer [106].

**Diabetes:** PDAC is more frequent in diabetic individuals. Type 2 diabetes mellitus has been increasing in recent years, and some authors believe that this is one of the causes of the increase in pancreatic cancer [107]. The risk is increased by 1.5 to 2 fold [108]. However, the relationship between diabetes type 2 and PDAC is a complex and non-linear one. The fact that many patients with PC are also diabetic [109] is not clear evidence of causality. Furthermore, there are conflicting statistics regarding diabetes prevalence in PDAC patients. Noy and Bilezikian reported a wide range of prevalence that goes from 4% to reach 20% [110]. Importantly, these researchers identified a form of diabetes with specific hallmarks (they call it atypical diabetes) that preceded PDAC and could help to identify early-stage PDAC. It is characterized by:(a)brief history of diabetes;(b)lack of family history;(c)lack of obesity;(d)rapid progression to insulin dependence.

Diabetes treatment also plays a role in PC risk. Bodmer et al. [111] found that diabetics treated with metformin had a low risk of PC (OR 0.87), but those receiving sulfonylureas (OR 1.9) or insulin (OR 2.29) had a markedly increased risk.

**Families with hereditary cancer predisposition syndrome:** in addition to familial cancer, there is a group of hereditary mutations that predispose patients to cancer in general and for certain types of cancer in particular. Some of these syndromes also predispose patients to pancreatic cancer. Table 1 shows a list of these cancer predisposition syndromes. Those with a higher risk of pancreatic cancer are highlighted in yellow.

By age 70, one out of every four persons with the Peutz–Jeghers syndrome has developed pancreatic cancer (at an average age of 55) [118]. The TP53 mutation in the Li Fraumeni syndrome showed a 7.7 relative risk for pancreatic cancer [119]. These examples show that the risk is much higher in the highlighted diseases, but there are important differences among them. Peutz–Jeggers syndrome has the highest cumulative rate of pancreatic cancer when compared with other hereditary cancer predisposition syndromes.

Von Hippel–Lindau disease usually produces pancreatic anomalies, mainly cysts (15% incidence), but they do not progress towards malignancy. Endocrine pancreatic tumors are also frequent [120].

**Unexpected weight loss (UWL) (decrease in body weight of 5% or more).** In a retrospective cohort study, 63,973 patients with UWL 1375 (2.2%) were found to have cancer within 2 years. On average, the patients were diagnosed with cancer after 180 days. PDAC was found in 5.3% of male cancer patients and 5.8% in females [121].

Although UWL is considered a late symptom of pancreatic cancer [122,123], it is found in 10% of cases and has a high predictive value among PDAC symptoms (it is only inferior to jaundice) [124].

Even though it appears late in the disease, there is evidence that UWL frequently precedes other cancer symptoms [125].

According to the British Society of Gastroenterology [126], UWL is one of the causes for urgent referral of the patient for a full workup regarding PDAC.

Based on the above risk factors, a mathematical simulation model was developed for risk evaluation that makes it possible to grade the risk of any individual case. Results above 50 points permit the inclusion of a patient in the high-risk group. Patients with a score above 50 represent an OR above 2.5 or higher. The lower limit, or cutoff level, needs to be accurately determined through prospective studies. Unfortunately, many retrospective population studies lack all the data used to build the scoring system.

The scoring system is shown in Table 2.

This scoring system makes it possible to distinguish four levels of risk:(a)low: score under 30(b)intermediate: score between 31 and 50(c)high: score between 51 and 75(d)very high: score above 75

This scoring system should be handled by the general practitioner. If the score is above 50, the patient should be referred to a specialist.

## 9. Part 2: High-Risk Population Screening

### 9.1. Biochemical Screening

There is no recognized early biochemical marker of pancreatic cancer. The classic CA19-9 is usually increased in pancreatic cancer, but there is no evidence that this is an early event. Something similar occurs with other biochemical markers such as carcinoembrionic antigen (CEA), S600A6, and osteopontin (reviewed by Li [161]). Osteopontin is probably one of the earliest markers.

Through an extensive review of the literature, we found no early biochemical marker of PC. Therefore, the main problem with all the available markers is not so much their specificity and their sensitivity but their “lateness”. We believe that the available biomarkers are not very useful for early diagnosis.

**CA19-9 (CARBOHYDRATE ANTIGEN 19-9):** this is the only serum biomarker approved by the FDA (United States Food and Drug Administration) for diagnostic purposes [162]. CA19-9 is an antigenic protein that can be defined by monoclonal antibody binding to CA19-9, the tumor surface marker Sialyl-Lewis A. (CA19-9 is a sialylated Lewis blood group antigen). It was discovered in 1979 in colorectal carcinoma and later in pancreatic cancer [163,164].

Guidelines from the American Society of Clinical Oncology discourage the use of CA19-9 as a screening test for pancreatic cancer. This is due to the lack of ability to detect asymptomatic PCs [165,166].

Sensitivity is approximately 80%, and specificity is around 65% [167,168,169]. The diagnostic sensitivity and specificity of CA19-9 measurement are not high enough to be used in the early stage of PC. Only 65% of patients with a resectable tumor showed increased CA19-9 [170]. As a screening tool, CA19-9 associated with diagnostic imaging may be helpful in patients with risk scores between 30 and 75. However, many patients with small PC will go undiagnosed. Kim et al. [171] screened over 70,000 persons between 1994 and 2000 and found that CA19-9 is ineffective for screening asymptomatic populations. In this regard, adding MRI or CT scan to CA19-9 will improve diagnostic accuracy, though it depends fully on the imaging.

**Osteopontin (OPN)** is a secreted glycophosphoprotein which is the product of tumor infiltrating macrophages but not of pancreatic tumor cells. Koopman et al. [172] found that OPN was increased in the serum of all 50 patients with early diagnosis treated surgically and in none of the normal controls. They used a cut-off level of 330 ng/mL and ELISA measurement. They also established that “elevated OPN had a sensitivity of 80% and specificity of 97% for pancreatic cancer. In contrast, only 62% of these patients with resectable pancreatic cancer had elevated CA19-9”.

One of the problems with OPN is that it is also increased in chronic pancreatitis, thus impeding differentiation between cancer and pancreatic inflammation [173]. On this point, there are some controversies: Rychlikova et al. [174] maintain that an OPN level above 102 ng/mL is diagnostic for PC. The problem is that it seems that such a high level of OPN is usually found in advanced stages. Our opinion is that OPN is not a useful marker for early diagnosis of PC in patients with a history of chronic pancreatitis.

In a meta-analysis of 491 PC patients and 481 healthy controls, OPN was found to be significantly increased in early-stage PC [161]. However, that paper does not mention if there were patients with a history of chronic pancreatitis.

Poruk et al. [175] tested the serum of 86 patients with early PC for OPN and TIMP-1 (tissue inhibitor of metalloproteases-1) and concluded that both markers were useful for the early diagnosis of PC and allowed the exclusion of chronic pancreatitis.

**TIMP-1 (tissue inhibitor of metalloproteases-1):** Tissue inhibitors of metalloproteinases (TIMPs) are endogenous protein regulators of the matrix metalloproteinase (MMPs) and ADAMs families [176]. One of these inhibitors is overexpressed in pancreatic cancer, and its serum level is also increased. Its value as a diagnostic tumor marker is lower than that of CA19-9 regarding sensitivity and specificity [177]. It did not improve diagnostic accuracy when added to CA19-9. However, increased TIMP-1 urinary levels in patients with PC allowed the discrimination of healthy controls from patients with PC [178]. Unfortunately, the authors do not mention the tumor stage in which TIMP-1 is increased in urine.

**MMP2 (MATRIX METALLOPROTEASE 2):** is an enzyme that degrades collagen type IV. MMP2 is highly expressed in PC cells, including its stroma [179].

**MUC4:** MUC4 is a high molecular weight glycoprotein that is over-expressed in pancreatic cancer tissues but not in pancreatic inflammatory diseases [180,181]. Importantly, it can be detected in plasma as well. MUC4 expression increases progressively in advancing states of PC [182].

**MUC5AC:** is a glycosylated, high-molecular-weight glycosylated protein expressed quite early in precancerous pancreatic cells [183]. It may be useful for the pathologist to determine borderline cells, but it is not a serum protein with diagnostic potential.

**TPS (tissue polypeptide antigen specific):** TPS is a specific epitope of the c-terminal part of human cytokeratin 18. Nine patients with symptomatic PC showed a TPS level above 100 U/L, while the non-oncologic controls had a level below 80 U/L [184]. Unfortunately, TPS increase is not an early phenomenon in PC because 267 pre-diagnostic PDAC plasma samples obtained years before clinical PDAC diagnosis did not show any rise in TPS [185].

**S600A6:** this is a protein of the S600 family that binds Zn[2]^+^, Ca[2]^+^, and Cu[2]^+^ and participates in the regulation of diverse cell functions, many of which are involved in tumor progression. Importantly, S600A6 was found to be over-expressed in pancreatic cancer [186]. Ohuchida et al. [187] measured mRNA S600A6 levels in the pancreatic juice of normal and pancreatic cancer individuals. All the cases with cancer were found to have an increased level compared with non-neoplastic patients, including those with chronic pancreatitis. Furthermore, S600A6 was increased in the very early phases of pancreatic carcinogenesis [188].

**PC-594:** is the result of a metabolomic search for a biomarker [189]. PC-594 is a circulating 36-carbon polyunsaturated fatty acid that can be identified in normal serum through mass spectrometry. This fatty acid is significantly decreased in pancreatic cancer (0.76 ± 0.07 µmol/L versus 2.79 ± 0.15 µmol/L in control subjects) [190]. Sensitivity was 90%, and specificity was 87%. These findings were corroborated by further research [191]. Despite the seemingly reliable results, we found no further studies with PC-594 as a pancreatic cancer marker.

**MIC-1 (macrophage inhibitory cytokine-1)**: Koopman et al. [192] found that this cytokine was an early marker that allowed differentiating resectable from non-resectable tumors. Furthermore, it was an independent biomarker not related to CA19-9. They also found that: “*MIC-1 was significantly better than CA19-9 in differentiating patients with pancreatic cancer from healthy controls, but not in distinguishing pancreatic cancer from chronic pancreatitis*”.

We believe that there are promising early markers that can help establish a battery of tests with high sensitivity. For example, a battery made up of CA19-9, OPN, TIMP-1, MUC 4, MMP2, MIC-1 and PC-549 can provide a reasonable base for determining the high-risk population that needs to go further with image screening and, at the same time, limiting the number of cases that need such studies. This needs experimental testing with well- planned prospective clinical trials.

However, the association of different markers has not yielded significant improvement in early diagnosis or in screening accuracy as yet. Unfortunately, there is a lack of clinical trials on the multi-markers associations.

**Hepatocyte growth factor (HGF):** HGF is a protein related to the main hallmarks of cancer, participating in proliferation, migration, angiogenesis and drug resistance. It is frequently over-expressed in PDAC tissue samples. Importantly it is increate 10-fold in the serum of patients with PDAC compared with normal controls [193]. However, there is insufficient evidence that increased serum HGF increase is an early marker. Its receptor, c-Met, is also over-expressed in pancreatic cancer cells [194].

**Inflammatory markers** are frequently increased in PDAC and are related to pro-tumoral promotion. They are absolutely unspecific, and their plasma increase only reflects the presence of an inflammatory process, whether of tumoral or non-tumoral origin.

The pancreatic cancer microenvironment is highly inflammatory and participates in the creation of an anergic milieu which is partly responsible for resistance to treatments [195].

**Fibrinogen:** While albumin synthesis was not found to be decreased, fibrinogen synthesis was increased many fold in PDAC patients with cachexia [196]. Inflammatory markers, such as C reactive protein and fibrinogen were increased in almost all the patients with advanced PDAC [197]. The fibrinogen-to-albumin ratio correlates with the progression of the tumor [198], but this is a prognostic rather than a diagnostic biomarker [199]. Serum fibrinogen degradation products are also increased in pancreatic cancer, but they are not more sensitive than CA19-9 [200].

**C-reactive protein (CRP)**: CRP, an important inflammatory marker, has a predictive value in PDAC progression, but it is not a diagnostic marker [201,202] or an early increased marker. In two-nested case-control studies of serum C-reactive protein, Douglas et al. [203] found no support for the hypothesis that higher CRP concentrations were associated with incident pancreatic cancer. Thus, CRP is not useful for screening purposes.

**Interleukin-6 (IL6):** This inflammatory marker is increased in chronic pancreatitis and PDAC, but importantly this increase is significantly higher in PDAC [204].

However, as reliable as these inflammatory markers may be for advanced tumors, we are mainly concerned with early markers, and in this regard, they offer no advantages for early diagnosis or for screening. On the other hand, inflammatory markers can be helpful in the differential diagnosis between chronic pancreatitis and pancreatic cancer and as prognostic markers [205,206,207] (Table 3).

### 9.2. Image Screening

Although imaging may seem an infallible and perfect diagnostic method for the detection of early tumors, this is not so. In the first place, it is fallible because very small tumors are difficult to find by methods such as computed tomography, magnetic resonance cholangiopancreatography and ultrasound studies. By the time the tumors are visible, they are not so small.

Imaging and, eventually, endoscopy are very useful for determining unresectability. However, they are not very effective in diagnosing a small tumor of a few millimeters or an intraductal neoplasia that can be treated surgically.

The other problem is that it is difficult to distinguish small benign lesions from early malignant developments. Although somewhat more accurate, endoscopic ultrasound is a more complex and costly procedure. Endoscopic ultrasound-guided fine-needle aspiration biopsy can be considered the standard procedure for diagnosis in case of doubts. The problem is that this intervention is only feasible after a suspicious image shows up in some other image-based diagnostic procedure. Furthermore, it is a step beyond any screening method for large populations. 

## 10. Screening Populations with a Very-High-Risk Level (Score above 75)

Pancreatic cancer has a low incidence in the general population; therefore, indiscriminate general screening is not an option. On the other hand, effective screening methods should be used in very high-risk populations.

This group of patients, mainly individuals from families with frequent pancreatic cancer, or with known cystic lesions, or hereditary cancer predisposition syndromes, are screened/followed up in first-world countries with:(a)MRI or MRI cholangiopancreatography and(b)Endoscopic ultrasound (EUS) with or without fine needle aspiration/biopsy

Diverse schemes can be followed. A frequently used screening program can consist of magnetic resonance imaging once a year, followed by additional investigations if there are abnormal findings [208]. With this approach and with a follow up of 262 patients for a little over four years, three pancreatic cancers were detected. In one case, the tumor recurred after surgery and the other two developed metastasis. At first impression, this scheme did not detect malignancies early enough.

Another scheme used CT scan and EUS. If EUS showed alterations, fine needle aspiration or biopsy and endoscopic retrograde cholangiopancreatography were performed [209]. This method allowed the identification of eight patients with pancreatic cancer in a population of 78 high-risk patients. This means that the follow up identified 10% of an asymptomatic high-risk population as harboring a malignant tumor.

Langer et al. [210] followed 76 high-risk individuals for over 5 years with MRI, EUS, and eventual fine needle aspiration. Seven patients underwent surgery for suspected lesions. They found no pancreatic cancer but mainly intraepithelial neoplasias and intraductal tumors. Considering that these are preneoplastic lesions, we may consider that pancreatic cancer was prevented in six of the seven operated cases.

Poley et al. [211] studied 44 high-risk individuals (hereditary cancer-predisposing syndromes and familial pancreatic cancer) with EUS. These patients had never been screened and were asymptomatic. They found that three had asymptomatic pancreatic mass lesions that were successfully removed. Sizes were 12, 27 and 50 mm. Additionally, they found seven patients harboring intraductal papillary mucinous neoplasia. EUS revealed the presence of 7% of cancers and 16% of precancerous lesions in this high-risk population.

At this point, we can consider EUS and EUS-guided fine needle aspiration (EUS-FNA) or EUS-guided fine needle biopsy (EUS-FNB) as the most effective methods for PC screening in high-risk populations. EUS, EUS-FNA, and EUS-FNB are highly accurate for detection and diagnosis [212]. It was not possible to establish EUS superiority over other imaging methods, such as CT or MRI, in the initial stage of use because all have their own limitations [213], and we believe that EUS and MRI or CT should be used in a complementary manner. However, nowadays, EUS with or without needle aspiration/biopsy can be considered superior to CT for detecting early pancreatic lesions. Gonzalo Marin et al. [214] summarized this concept with the following words: “*EUS has proved rates higher than 90%, especially for lesions less than 2–3 cm in size in which it reaches a sensitivity rate of 99% vs. 55% for CT. Besides, EUS has a very high negative predictive value, and thus EUS can reliably exclude pancreatic cancer*”.

This superiority of EUS over other procedures for early diagnosis has been confirmed by many authors [215,216]. Another advantage is that tissue samples can be obtained during the procedure [217].

Puli et al. [218] found that EUS-FNA had a sensitivity of 86.8% and a specificity of 95.8%. These results are superior to any other diagnostic method.

Endoscopic ultrasound diagnostic efficiency has been enriched in the last years with the addition of endoscopic contrast agents, which led to improved imaging [219,220,221,222].

The endoscopic ultrasound fine needle aspiration (EUSA), since the advent of better fine needles for biopsy, has been evolving towards endoscopic ultrasound fine needle biopsy (EUSB) obtaining tissue samples instead of cytology studies without increasing complications [223]. Increasing imaging studies of the abdomen resulted in the increased finding of asymptomatic pancreatic cystic lesions. Some of these lesions may harbor malignant cells, and therefore cystic fluid analysis or biopsy is required [224,225,226,227,228,229,230,231,232,233,234,235,236,237,238,239,240,241,242,243,244,245,246,247,248,249,250,251,252,253,254,255,256,257,258,259,260,261,262,263,264,265,266,267].

However, EUSB in pancreatic cysts is not devoid of adverse events [228].

## 11. Liquid Biopsy (LB) for Pancreatic Cancer Screening

Malignant tumors and also their metastases can release cells and parts of the cells that can be found in blood and other biological fluids, which can be useful for diagnostic purposes. Among these released materials, it is possible to find circulating tumor cells (CTCs), circulating cell-free nucleic acids (cfDNA and cfRNAs), and circulating extracellular vesicles: exosomes.

cfDNA is increased in cancer patients in general [229,230] and in PDAC in particular [231,232].

cfDNAs reach the blood stream through cell turnover or cell death [233].

Methodology and results were recently reviewed by Heredia-Soto et al. [234] and Kumar Yadav et al. [235]. Although interest in the subject seems to be a new development, Mandel and Metais identified circulating DNA in 1948 [236,237]. But the actual initiators of its research and development for oncological purposes were Sorensen et al. [238].

They have not been systematically introduced in clinical practice regarding pancreatic cancer as yet. The analysis of these materials requires sequencing or other molecular methods to identify specific mutations that can be found in pancreatic cancer. These mutations usually are in the *KRAS*, *CDKN2A*, *TP53* and *SMAD4* genes. However, these mutations can also be found in non-malignant diseases or low-grade intraductal neoplasias [239].

Kras mutations are found in normal pancreas, in chronic pancreatitis with ductal hyperplasia and in pancreatic adenocarcinoma. According to Tada et al., Kras mutation occurs frequently in hyperplastic foci in pancreatic ducts [240].

A meta-analysis of 2156 patients concluded that the detection of Kras mutations in pancreatic exocrine secretions did not provide sufficient specificity nor sensitivity to distinguish PDAC patients from chronic pancreatitis or pre-malignant lesions, or healthy individuals [241].

## 12. Promoter Methylation Status of Genes in cfDNA

**ADAMTS1 and BNC1.** ADAMTS1 is a disintegrin and metalloproteinase with thrombospondin motifs 1 protein; BNC1 is the gene that codes for the zinc finger protein basonuclin-1. In 2019, Eissa et al. [242] showed that the methylation status of two genes in cell-free DNA, ADAMTS1 and BNC1, was highly sensitive and specific for early PC diagnosis.

**SPARC (Secreted Protein Acidic and Rich in Cysteine), UCHL1 (ubiquitin carboxy-terminal hydrolase L1), PENK (proenkephalin), and NPTX2 (neuronal pentraxin 2)** are four genes that Sing et al. [243] have found with methylated promoter regions in cfDNA in patients with pancreatic cancer. SPARC has been shown to be an early marker that permits the differential diagnosis between PC and chronic pancreatitis. Simultaneous high mutilation of SPARC and NPTX2 was found in metastasized PC. UCHL1 methylation correlated with advanced disease. In conclusion, SPARC is an early marker as long as the other genes are not promoter methylated.

Accumulated evidence shows that the methylation profiles of certain genes in cfDNA are useful markers for distinguishing between chronic pancreatitis and cancer [244], and in certain cases, such as SPARC, may be an early marker of the disease.

After a thorough search of published reports, we concluded that most of the possible markers that can be identified through liquid biopsy have a prognostic rather than a diagnostic utility. Most of the patients studied with mutation-based liquid biopsies were found to have advanced-stage PC [245].

Those that may be useful for early diagnosis are:
♦Promoter methylation of BNC1 and ADAMTS1.♦Small mutant fragments of cfDNA [246].♦According to Berger et al. [247], the total amount of cfDNA may discriminate between patients having early pancreatic cancer or a preinvasive lesion and normal subjects.♦CancerSEEK is a screening multi-test that studies ctDNA (16 genes including *KRAS*) and eight cancer-associated proteins in blood: (carbohydrate antigen 125 (CA-125), CA19-9, CEA, HGF (hepatic growth factor), myeloperoxidase, prolactin, OPN, tissue inhibitor of metalloproteinases 1 (TIMP-1)). CancerSEEK was developed by Cohen et al. [245] as a screening system for different tumors (ovary, liver, esophagus, pancreas, gastric, colorectal, lung, and breast cancer) at relatively early stages. The sensitivity was 70% for PC, and the specificity was high [248].

## 13. Circulating Exosomes

We believe that circulating exosomes have the highest potential for PC screening because [249]:(1)They are permanently shed from normal and tumor cells.(2)Shedding from tumor cells is more copious.(3)Identifying the message they are carrying can be diagnostic for PC.(4)No interventional procedure beyond blood sampling is involved.(5)They can rule out inflammatory diseases of the pancreas.(6)The genetic signature of the cancer can be obtained in many cases.(7)Experimental evidence had shown a very high sensitivity, close to 100%.(8)Exosome release is increased in malignant cells [250,251].

The main drawback is that a somewhat sophisticated laboratory is needed.

Exosomes are simple membrane particles generated through the endolysosomal pathway. They are regularly produced by normal and cancer cells and released into the extracellular matrix from where they enter the circulatory system. They represent a complex mechanism of intercellular communication. Exosomes carry instructions in the form of nucleic acids, proteins (including immunoinhibitory proteins [252]), and metabolites. The nucleic acids are DNA, messenger RNA microRNA and long non-coding RNA [253,254,255]. Kharaziha et al. [256] described tumor-derived exosomes in a very vivid way: a message in a bottle.

In addition to the multiple effects of exosomes in pancreatic cancer progression [257,258,259] that we shall not discuss here, exosomes can be used for diagnostic purposes.

Exosomes are quite stable in biological fluids. Thus, harvesting them is not a complex issue [260]. The importance of exosomes for diagnostic purposes lies in their content, which represents the message they are carrying. Two miRNAs, -21 and -221, are particularly over-expressed in pancreatic cancer tissues [261]. Down-regulation of these two miRNAs decreased migration/invasion and reduced the expression of NF-kB and Kras.

Exosomal miRNA 21 is usually found to be increased in PC with a sensitivity of 95.5% and specificity of 81.5% [262].

MiRNA17-5p was also found to be increased in circulating exosomes of patients with pancreatic cancer [263], but this seems to occur with a more important tumor burden. Thus, it is not an early event.

## 14. Glypican-1 (GPC1) in Exosomes

In 2015, Melo et al. [264] published an article which established that exosomes of PC patients had a higher expression of glypican-1 compared with normal controls. This report tested 251 patients with PC.

According to the authors, “GPC1^+^ exosomes were detected in the serum of patients with pancreas cancer with absolute specificity and sensitivity, distinguishing healthy subjects and patients with a benign pancreas disease from patients with early and late stage pancreas cancer”.

Furthermore, the level of exosomes +glypican correlated with the tumor burden and disease progression.

Glypican-1 is a membrane-anchored protein that is increased not only in pancreatic cancer but in other tumor tissues as well [265,266]. It was found to be increased in circulating exosomes of colorectal cancer patients [267].

The method found by Melo et al. [262] seems to be the best diagnostic tool for early PC diagnosis. However, it is not simple to be introduced in daily practice and even less in large population screening because it is quite difficult to isolate glypican-positive exosomes by flow cytometry [268]. Frampton et al. [269] confirmed the increased level of glypican-1 positive exosomes in pancreatic cancer in a small number of patients. There are no publications about exosomal glypican-1 as a diagnostic resource in PC after those by Melo et al. and Frampton et al. Thus, we think that the method requires further validation with large populations. Another reasonable doubt is the specificity of glypican-1 positive exosomes because they can be found in other tumors as well.

Glypican-1 enrichment of exosomes seems to be related to hypoxia. Heparin can have a similar effect [268,270]. Zhao et al. found that exosomal glypican-1 enrichment is associated with early recurrence of pancreatic cancer [271]. Additionally, they found that exosomal glypican-1 was a useful method for distinguishing early PC from healthy controls but not from chronic pancreatitis.

Lai et al. [272] maintain that a microRNA signature in circulating exosomes is superior to glypican-1 for diagnostic purposes. According to these authors, “high exosomal levels of microRNA-10b, (miR-10b), miR-21, miR-30c, and miR-181a and low miR-let7a readily differentiate PDAC from normal control and CP samples”. In addition to this signature, Slater et al. [273] reported that miR196a and miR196b are early biomarkers of familial pancreatic cancer.

In summary, the problem with exosomal glypican-1 is that it does not differentiate pancreatic cancer from benign pancreatic diseases [274].

For additional information on exosomal glypican-1 in early diagnosis of pancreatic cancer, see references [275,276,277,278] (Figure 2).

Figure 3 shows the evolutionary phases of pancreatic cancer.

Despite some controversies, we believe that exosomal glypican-1 is an early marker of pancreatic cancer that can be useful when included in a battery of tests rather than an isolated study.

This chapter on molecular diagnostics is in active development, and it will probably lead to a breakthrough in the near future.

## 15. Endoscopic Ultrasound (EUS)

EUS is an interventional outpatient procedure in which the endoscope is introduced through the mouth, reaching the duodenum and performing an ultrasound mapping of the pancreas. It requires sedation or mild anesthesia. EUS is used to find small pancreatic lesions that cannot be detected by CT or MRI scans [282]. Pancreatic intraepithelial neoplasias can be detected by EUS but not by other imaging procedures. These very early lesions have the potential to progress towards an invasive adenocarcinoma within a few years [283]. One advantage of EUS is that tissue samples or cells can be obtained by EUS-FB or EUS-FA.

## 16. Is Acute Pancreatitis an Early Marker of Pancreatic Cancer?

Two cohort studies, one from Denmark and another from the US, showed in the first case that 1.4% of cases of PDAC had acute pancreatitis before cancer. This proportion increased to 5.9% in the US cohort [284]. Importantly, patients with pancreatic cancer that had acute pancreatitis up to 90 days prior to diagnosis had an earlier-stage tumor and a better outcome. The possible explanation is that acute pancreatitis may be produced by tumor obstruction of pancreatic ducts and enzymatic release, thus driving attention towards the pancreas at an earlier stage.

## 17. Discussion

In 2019 the US Preventive Services Task Force issued a recommendation against pancreatic cancer screening [285] based on the fact that “approximately 1.6% of individuals in the United States will develop pancreatic cancer during their lifetime. With this relatively low prevalence, even an ideal screening test with 99% sensitivity and 99% specificity would yield 1000 false-positive results if applied to 100,000 patients” [286]. However, if the group of high-risk patients can be clearly identified, screening may become a fundamental tool for an earlier diagnosis.

Early diagnosis is considered a fundamental condition for a better outcome in all cancers [287,288,289]. Pancreatic cancer is no exception [290].

The main reason for therapeutic failure in pancreatic cancer is late diagnosis, despite the 15 years it takes for the tumor to reach full development. Lack of symptoms or warning signs until late is the cause of this delay. Currently, there are no agreed or conventional PC screening programs. Importantly, there are no programs directed towards early diagnosis. The main reason for this is the lack of reliable markers. The only officially accepted marker is CA19-9, and it is not very effective for early diagnosis.

Although all the publications show that PC has a nefarious prognosis, when tumors are less than 10 mm the 5-year survival is approximately 60% [291,292]. Therefore, early diagnosis is currently the only positive approach to this disease.

Early diagnosis requires reliable screening methodologies. Unfortunately, these methodologies have not yet been developed. This also requires a clear identification of the population at risk [293].

Interestingly, in its Clinical Guidelines version 2.2017, the National Comprehensive Cancer Network [294] does not dedicate a single paragraph to pre-symptomatic stage diagnosis. The only, indirect mention is “*The best-validated and most clinically useful biomarker for early detection and surveillance of pancreatic cancer is CA19-9, a sialylated Lewis A blood group antigen. CA19-9 is a good diagnostic marker, with sensitivity of 79% to 81% and specificity of 80% to 90% in symptomatic patients, but its low positive predictive value makes it a poor biomarker for screening*”.

Large population screenings with CA19-9 and non-endoscopic ultrasound have been shown to be ineffective for PC diagnosis in asymptomatic individuals [295].

However, pancreatic precursor lesions have been known for more than a hundred years [296], and we now know that it takes many years for these lesions to develop into an invasive tumor.

General population pancreatic cancer screening has not been recommended because of the low incidence of the disease [67]. We can partially agree with this. However, this conduct leaves out a great part of the population that does not have a very high risk for PC (score above 75) but undoubtedly has a higher risk compared with the general population.

As an example, the incidence of pancreatic cancer in the general population (13/100,000) increases to 68/100,000 after age 55. Although we have no data, we are convinced that it further increases after age 65.

It is the group of individuals with a score between 50 and 75 that we believe should be screened. Importantly, we should not fall prey to the nihilistic idea that it is not worth the cost and effort to screen populations at higher risk but not in the very high bracket. Furthermore, the incidence of pancreatic cancer is rising, although the reasons are not clear.

It is worth the effort to find an efficient screening method for the population at risk scoring between 50 and 75. The best example to justify this effort is that Ariyama et al. [297,298] found excellent survival rates in 79 patients with tumors of less than 1 cm who underwent surgical resection.

There is only one way to modify this situation: identifying the population at risk and screening them for the early signs of the disease according to their score. Indiscriminate population screening is not an option because it involves a high cost and more than a few inconveniences for the individuals. Furthermore, the incidence of the disease, although rising constantly, is not that high (13 cases per 100,000). Thus the cost/benefit ratio is very unbalanced.

It is very simple to identify the high-risk population among individuals belonging to families with a high incidence of pancreatic cancer where a hereditary factor is usually involved. It is equally simple to identify those harboring a hereditary predisposition to cancer syndromes, such as Peutz–Jeghers, Li Fraumeni, or Lynch, among others. Things become more complicated in populations lacking these predisposing factors.

Based on extensively published risk factors for pancreatic cancer, we present here a scoring table that can help identify a group of individuals with a score between 50 and 75 that deserve to be screened periodically. In this scoring table, all the hereditary cancer-predisposing diseases are above 75.

The screening methodology above 75 and in the 50–75 range should differ in such a way as to reduce the cost and inconveniences in the latter group.

While those with a score above 75 need MRI or CT scan imaging plus EUS, those in the 50–75 range should be studied with imaging and a battery of serum biomarkers.

Unfortunately, serum biomarkers are not early hallmarks of PC, but combining at least three of them, such as CA19-9, OPN, and MUC4, with MRI will identify a subset of patients that should undergo EUS. This battery can be complemented with exosomal studies, which seem to be more reliable for early identification.

Some authors maintain that screening for pancreatic cancer would prove to have very few benefits [299]. In this paper, we have discussed the evidence that contradicts this thinking. The evidence in favor of screening is blatant for the very high-risk group (score above 75). The next two groups may be a matter for further discussion. In this regard, we propose here a scoring mechanism that defines the population to be screened as well as the screening approach (Figure 4).

Understanding the long development of pancreatic tumors, which require more than 15 years, gives plenty of opportunities for detection before they become inoperable.

Another issue to consider is the problem of procrastination produced by the patient’s delay in seeking consultation, delay due to health system bureaucracy, or delay attributed to the physician. In this last case, Moosa et al. [300] frequently found “reluctance to refer the patient immediately to an established diagnostic center”. In our experience, between the first symptoms and the surgery there was a gap of approximately three months. Considering that at the time of the first symptoms, the tumor has an average diameter of 3 cm, knowing the tumor volume duplication time, we can calculate that the surgeon will find a tumor of at least 4.5 cm. after a 3-month delay. Elective cancer surgery is not urgent in most cases. Pancreatic cancer, on the other hand, should be considered almost urgent. Both the physicians involved in the treatment and the healthcare system should take measures to have the patient operated on as soon as possible, meaning not more than a 3-week delay.

In 2012 and 2013 Hippisley-Cox and Coupland [301,302,303] published three articles to help primary care physicians identify the risk of pancreatic cancer in patients before clear symptoms develop. The signs and symptoms mentioned in these publications point to an established pancreatic cancer a short time before the cardinal symptoms appear. Based on the precancer signs and symptoms of a large population, these researchers established an algorithm that scores pancreatic cancer risk.

Beyond the important practical implications of these publications, we have doubts about how early the diagnosis can be achieved through the algorithm. The elements included in it seem to identify cancer in an invasive rather than a pre-invasive stage.

The risk calculations proposed by Hippisley-Cox and Coupland may be integrated into the algorithm proposed in this paper in order to achieve an earlier diagnosis.

## 18. Conclusions

PDAC is one of the most fatal malignancies, which has been only minimally modified in the last 30 years. It is difficult to screen PDAC in the general population. This paper described a method to reduce the screened population to selected high-risk individuals and also reviewed the available screening methods.

Regarding screening methods, the following conclusions were reached:(1)Serum biomarkers appear too late in pancreatic ductal adenocarcinoma and are therefore ineffective for timely diagnosis. Despite this limitation, using a battery of markers can probably identify tumors in a stage when they are still candidates for surgery.(2)Imaging alone with MRI does not allow early diagnosis, and it is not very effective for screening purposes.(3)The association of endoscopic ultrasound and CT scan or MRI imaging (and eventual fine needle aspiration when there are lesions) seems to yield the most reliable results and can contribute to early diagnosis of pancreatic malignancies.(4)Pancreatic juice study alone is insufficient for diagnosis, but it may represent an important complement to other methods.(5)Circulating cancer cells, free DNA and exosomes have not achieved clinical status yet, and more information is required in this regard. However, miRNAs in circulating exosomes can be useful for early diagnosis. Glypican-1 in exosomes require further testing.(6)For the time being, EUS associated with imaging studies is the most reliable screening method.

## Figures and Tables

**Figure 2 cancers-15-04430-f002:**
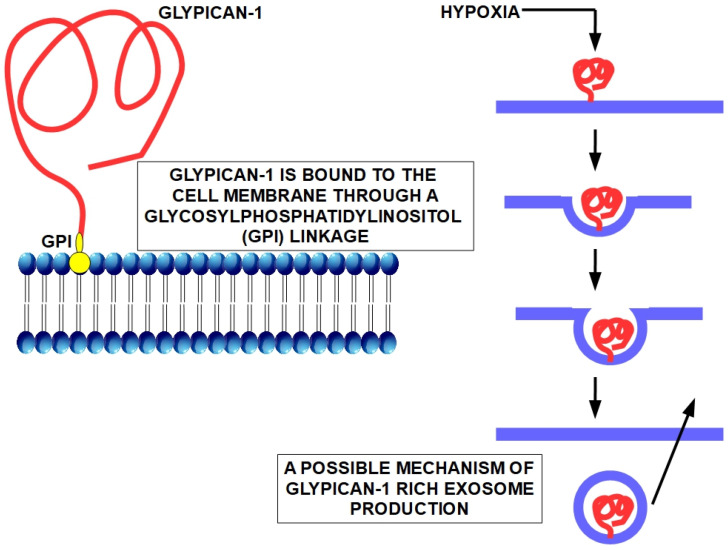
Left panel: structure of the protein glypican-1 linked to the cell membrane by glycosylphosphatidylinositol (GPI) (modified from Fico et al. [279]). Right panel: shows a possible mechanism of production of glypican-1-rich exosomes, although not experimentally confirmed. Hypoxia is one of the main factors in stimulating this production [280]. Pancreatic cancers have very high levels of hypoxia [281].

**Figure 3 cancers-15-04430-f003:**
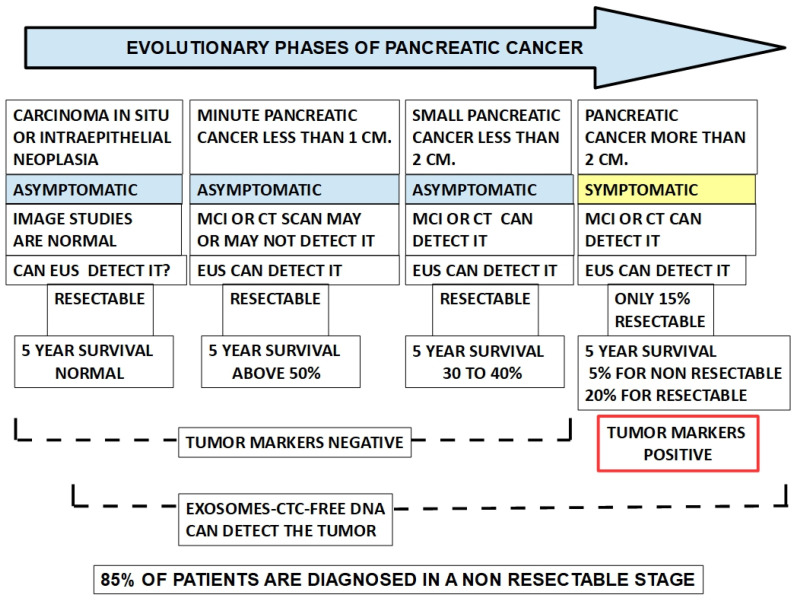
The diagram shows the correlation between tumor progression and diagnostic means. It is quite evident that tumor markers are not useful for early diagnosis, while EUS is the main contributor to achieving diagnosis in a period where the disease is still within the range of a possible surgical solution.

**Figure 4 cancers-15-04430-f004:**
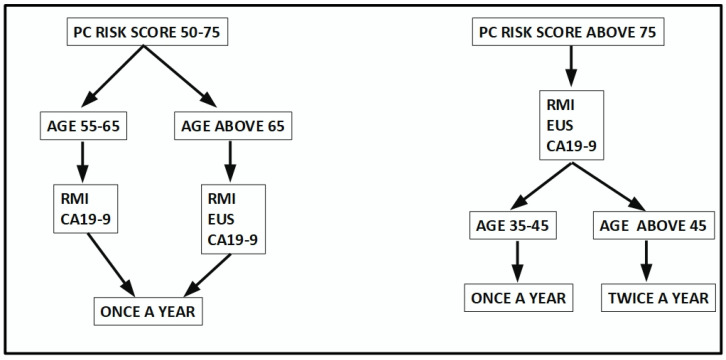
A proposed screening scheme for early diagnosis of pancreatic cancer based on the findings in the literature and personal experience.

**Table 1 cancers-15-04430-t001:** Hereditary cancer predisposition syndromes.

Disease	Higher Risk for Cancer of	Affected Gene
**BRCA1 MUTATION** **PDAC incidence 2.55%**	Breast, ovary, prostate, and pancreas	*BRCA1*
**BRCA2 MUTATION** **PDAC incidence 2.13** [112]	Breast, ovary, prostate, pancreas	*BRCA2*
Cowden syndrome	Breast, thyroid, endometrium	*PTEN*
Familial hereditary colon polyposis	Colorectal	*APC*
Hereditary diffuse gastric cancer	Diffuse gastric cancer, breast	*CDH1/CTNNA1*
Langerhans cell histiocytosis	Granulomas	*BRAF*
**Li-Fraumeni syndrome** **Accumulated PDAC incidence 33%**	Sarcomas, breast, brain, leukemia, pancreas	*TP53*
**Lynch syndrome** **Accumulated PDAC risk 3.7% and 8.6 fold increase** [113] **1.31% up to age 50 and 3.68 up to age 70** [114]	Colon, rectum, stomach, small intestine, liver, gall bladder, brain, prostate, upper urinary tract, pancreas	*MLH1*, *MSH2*, *MSH6*, *PMS2* or *EPCAM*
**Multiple endocrine neoplasia type 1**	Parathyroid, pituitary and pancreas	*MEN1*
Multiple endocrine neoplasia type 2 and 2 B	Medullary carcinoma of thyroid, pheochromocytoma	*RET*
**Familial atypical multiple mole melanoma (FAMMM)** [115]	Melanomas and moles Pancreatic cancer	Subset of patients with *CDKN2A* mutation
**Peutz Jeghers syndrome** [116] **Accumulated risk 33%**	Hamartomas and cancer in the digestive system, including the pancreas	*LKB1*
MYH-associated with polyposis	Colorectal cancer	*MYH*
Perlman syndrome	Wilms tumor	*DIS3L2*
**Von Hippel–Lindau disease** [117]	Hemangioblastomas, clear cell renal carcinoma, pancreatic cancer, neuroendocrine cancer, pheochromocytomas	*VHL*
Ataxia telangiectasia	Lymphoma and leukemia	*ATM*

**Table 2 cancers-15-04430-t002:** Pancreatic cancer risk evaluation.

‣ FAMILIAL HISTORY OF PANCREATIC CANCER	
CASE	
First-degree relative 2.33-fold risk increase [127]	20 points
Second-degree relative 1.28-fold risk increase [112]	5 points
2. CASES	
in first-degree relatives (6-fold increase) [128]	50 points
MORE THAN TWO CASES	
in first-degree relatives (30-fold increase) [113,129]	70 points
‣ HISTORY OF CHRONIC PANCREATITIS	35 points
OR 2.3 [130]/OR 2.2 with one discharge and OR 3.8 with multiple	
admissions [131]/OR 7.05 [132] (Asian population)	
‣ HISTORY OF ACUTE PANCREATITIS	10 points
OR: 2.07 [133]/OR 2.02 [134]	
‣ CIGARETTE SMOKING (OR 1.77–2.2) [135]	5 points
‣ More than 30 years smoking [136] (OR 2.4) adds	2 points
‣ More than one pack a day adds	2 points
‣ Black tobacco smoking adds	1 point
‣ AGE ABOVE 55 (general population)	10 points
‣ AGE ABOVE 65 (general population)	15 points
‣ AGE BELOW 55 (general population) subtract	−5 points
‣ AGE ABOVE 40 (hereditary cancer predisposition population)	15 points
‣ DIABETES WITHIN 3 YEARS OF ONSET (OR 3) [137]	10 points
‣ LONG-TERM DIABETES 2	7 points
‣ SUBCLINICAL DIABETES [138,139,140]	10 points
‣ DIABETES TREATED WITH INSULIN or SULFONYLUREAS Adds	5 points
‣ DIABETES TREATED WITH METFORMIN Subtract [141]	−2 points
‣ NON-ZERO BLOOD TYPE [142]	1 point
A (OR 1.32)	2 points
AB (OR 1.51)	
B (OR 1.72)	3 points
‣ BLACK RACE	15 points
‣ HIGH BODY MASS INDEX [143,144,145,146,147]	2 points
‣ HEREDITARY CANCER PREDISPOSITION SYNDROMES	60 points
OTHER THAN FAMILIAL PANCREATIC CANCER	
BRCA1/BRCA2, Li Fraumeni, Lynch, Peutz–Jeghers etc.	
‣ UNEXPECTED WEIGHT LOSS OF 5–9%	10 points
‣ LOSS ABOVE 9%	30 POINTS

Note 1 SMOKING CESSATION: 15 years after smoking cessation, risk returns to normal. Therefore, after 15 years, no points are added for ex-smokers. Note 2 ALCOHOL INTAKE: alcohol abuse was not included in the risk evaluation table because there are no consistent findings of significantly increased risk [148,149,150,151,152,153]. However, very high consumption of alcohol increases the risk [154,155] and should be awarded two points on the risk scale. Alcohol has indirect effects related to PC because it is a major cause of chronic pancreatitis. There is also a paradoxical finding showing an association between lower alcohol use from an early age and improved survival following pancreatic cancer [156]. This last situation does not add any information on risk. NOTE 3 SYNERGISTIC RISKS: It was found that cigarette smoking and familial history of PC were synergistic [157]. Therefore, when both risks are present, five points should be added to the total score. NOTE 4 AGE: less than 10% of cases of pancreatic cancer occur among people younger than 55. NOTE 5: If the individual has a Peutz–Jeggers syndrome, 10 points must be added to the final score. NOTE 6: if the patient has a hereditary cancer predisposition syndrome of the group highlighted in Table 1 and an age over 50, 15 points must be added to the final score instead of the ages considered for the general population. NOTE 7: We have not included dietary patterns among risk factors [158] because they are controversial issues, and quantification is unreliable. Furthermore, according to Michaud et al. [159], “dietary patterns were not associated with the risk of pancreatic cancer in two large cohort studies of men and women”. Ji et al. [160] found an inverse association between vegetable and fruit-based diets and pancreatic cancer. However, no positive associations were found.

**Table 3 cancers-15-04430-t003:** Usefulness of markers in early detection.

Marker	Detection
CA19-9	Usually does not detect asymptomatic cases. However, 65% of resectable cases had increased levels.
Osteopontin	It may be useful for early detection if chronic pancreatitis can be excluded.
Timp I	It may be useful for early detection, but sensitivity and specificity are low.
MUC 4	Its level increases with tumor growth; therefore, it has a low value for early detection.
TPS	It is not an early marker.
S600A6	It is an early marker when it is tested in pancreatic juice.
PC-594	Seems an early marker, but it has not been studied in depth.
MIC 1	Seems an early marker, but it has not been studied in depth.
HGF	There is no evidence that it is an early marker.
Inflammatory markers	Are not early markers.

## Data Availability

Data supporting this paper is originated from the authors listed in References.

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
