# Peer review of "Earlier Diagnosis of Pancreatic Cancer: Is It Possible?"

_cancers, 2023, doi:10.3390/cancers15184430_

Round 1

Reviewer 1 Report

I commend the Author for the huge job he did to gather and summarize all the information available on early diagnosis of pancreatic cancer.

I'd include a table reporting the performance of different screening methods

I suggest implementing the chapter on EUS by: 1) mentioning the EUS contrast technique, and its importance for the evaluation of pancreatic cancer precursors; 2) mentioning EUS-guided fine-needle biopsy, not only fine-needle aspiration (cite PMID: 35915956); 3) mentioning new tools for precise diagnosis of pancreatic cysts (cite PMID: 35451041)

Author Response

I commend the Author for the huge job he did to gather and summarize all the information available on early diagnosis of pancreatic cancer.

Thank you

I'd include a table reporting the performance of different screening methods

a table has been added

I suggest implementing the chapter on EUS by: 1) mentioning the EUS contrast technique, and its importance for the evaluation of pancreatic cancer precursors; 2) mentioning EUS-guided fine-needle biopsy, not only fine-needle aspiration (cite PMID: 35915956); 3) mentioning new tools for precise diagnosis of pancreatic cysts (cite PMID: 35451041)

All the suggestions have been added.

Thank you for your suggestions

Reviewer 2 Report

Astonishing is the lack of the current affiliation of the Author.The review contains many accurate clinical remarks, but generally do not provide much novelty.Those eternal problems with pancreatic cancer are known for many years.The Author wants to cover the wide range of the disease issues, which would be appropriate for monography and not review paper.Therefore some aspects are simplified or treated too superficially.Proposed scoring scale may be interesting but should be presented together with the clear guidelines. Who is supposed to perform the screening:general practicioners or tertiary reference centers? What are the indications, that combining the genetic and environmental risk factors will improve the screening efficacy?All the data up to now do not support the pancreatic screening cost effective even in high risk groups.

English needs minor editing by the native speaker

Author Response

Astonishing is the lack of the current affiliation of the Author.

The author does not have a current affiliation because he has recently retired after being the head of the surgical department and medical director of a 370 bed University affiliated Hospital in Buenos Aires for over 15 years. However, the reviewer is right. The author does not have an affiliation at the present moment.

At age 76 there are not many researchers that hold an affiliation.

The review contains many accurate clinical remarks, but generally do not provide much novelty. Those eternal problems with pancreatic cancer are known for many years.

A great part of the paper is a review. As that, its intention is not to introduce novelties. The fact that many of the problems are well known does not mean that they should not be summarized as an introduction to the central idea: that there are populations at high risk that should be identified and screened.

The Author wants to cover the wide range of the disease issues, which would be appropriate for monography and not review paper. Therefore some aspects are simplified or treated too superficially.

Unfortunately, this is an ample point. I cannot answer the objections unless the reviewer pinpoints which aspects were superficially discussed or over-simplified.

Proposed scoring scale may be interesting but should be presented together with the clear guidelines. Who is supposed to perform the screening: general practitioners or tertiary reference centers?

A clear guideline point has been added to the paper.

What are the indications, that combining the genetic and environmental risk factors will improve the screening efficacy?

The screening method proposed in the paper is based on the experience of 235 patients treated at the Hospital del Centro Gallego of Buenos Aires between the years 1995 to 2005. Thirty one of these patients were surgically treated by the author. The method was built by studying the risk factors in these 235 patients plus an extensive review of the medical literature.

All the data up to now do not support the pancreatic screening cost effective even in high risk groups.

Precisely, this is the main point that this paper wants to change. When the risk factors in the 235 patients were reviewed the author arrived to the conclusion that most of them would have been detected much earlier if there would be a screening method.

I have not found in the literature a serious effort to identify the population at risk and screen specifically this reduced population.

Round 2

Reviewer 2 Report

Generally the article should be abbreviated, focusing on the novel information and the proposed screening algorythm.The data presentation needs to be better organized, screening and early detection methods are not described separately. The informations provided in the manuscript are not current.K-ras mutation does not rule out chronic pancreatitis , this mutation is seen about 30% of chronic pancreatitis patients (page 2, last sentence)

Figure1-this is not current enough schema from 2005, much novelties were shown after this publication

Pancreatic cancer (PC) growth model was too simplified, only few mutations mentioned, while there is much more new informations on the genetics of pancreatic cancer.

Pancreatic associated diabetes is the very interesting problem and should be explained compared to PC-accompanying type 2 diabetes and other endocrine pancreatic function alterations.

The proposed screening scoring system is interesting but needs more explanations on the practical side.High risk individuals are screened anyway, even in the absence of additional parameters, but will not be operated unless pancreatic mass is detected and this is often too advanced for radical operation, even with the very strict observation.Is it proposed that all chronic pancreatitis, smokers wih diabetes will be all screened with CT, MRI and EUS with biopsy?What would be biopsied in the absence of pancreatic mass?Generally EUS with biopsy is not the screening method as suggested by the author, but is used in advanced PC in order to get the pancreatic tissue needed for chemotherapy.All resectable pancreatic masses are subjected to surgery without delay and further examination, based on all guidelines.Exosomes are also not the screening candidates, since this evaluation is rarely available, which is confirmed by the author.Therefore exosomes should be mentioned in paragraph on future early PC detection method or omitted. EUS is not the method to detect PanIN unfortunately as stated by the author.Figure 3 needs explanation.What is normal 5 years survival?What are the evidences, that Ca in situ detection and surgery will permanently cure PC?The recommendations should be more precise.Performing all :CT, MRI and EUS in all high risk patients seems not realistic.What is the evidence to use EUS additionally in patients over 65?

The number of literature positions, more than 300 is far too high for the review paper.In addition most of them are not coming from the last 5 years , some from more than twenty years.

The manuscript should be reviewed by the native speaker

Author Response

The article should be abbreviated

This article is in a great part a review. That means that it should include all the important and prevailing publications. This is not a mini review. Thus I think it has the right size.

Screening and early detection methods are treated under different headings. I do cannot imagine any other way of separating them.

With or without K-ras mutation screening methods do not change. I agree with the reviewer that this point deserves further analysis. Therefore a paragraph in this regard is included.

The basic elements included in Figure 1 have not changed. The objective of the figure is to show the small diagnostic window for PC and that has not changed in the last 20 years.

It is not said that all chronic pancreatitis patients who are smokers with diabetes should have a biopsy. Biopsy is limited to those cases in which EUS shows a dubious image.

K-ras mutation is mentioned but not discussed. It would take at least another 30 pages to discuss the issue. But a paragraph, as mentioned above is included.

The article does not discuss all the mutations that can be found in pancreatic cancer. That is out of the scope of the article. However, there will be included a reference to a review on the subject.

The reviewer is very interested in diabetes in pancreatic cancer. This item is discussed in the paper. Probably not with the depth the reviewer pretends. To enter ina thorough analysis of the issue would require another manuscript probably as long as the one we are presenting here.